# Proteomics and Metabolomics for Cystic Fibrosis Research

**DOI:** 10.3390/ijms21155439

**Published:** 2020-07-30

**Authors:** Nara Liessi, Nicoletta Pedemonte, Andrea Armirotti, Clarissa Braccia

**Affiliations:** 1Analytical Chemistry Lab, Istituto Italiano di Tecnologia, Via Morego 30, 16163 Genova, Italy; nara.liessi@iit.it; 2U.O.C. Genetica Medica, IRCCS Giannina Gaslini, Via Gerolamo Gaslini 5, 16147 Genova, Italy; nicoletta.pedemonte@unige.it; 3D3PharmaChemistry, Istituto Italiano di Tecnologia, Via Morego 30, 16163 Genova, Italy; clarissa.braccia@iit.it

**Keywords:** cystic fibrosis, proteomics, metabolomics, post-translational modifications, interactomics, biomarker discovery

## Abstract

The aim of this review article is to introduce the reader to the state-of-the-art of the contribution that proteomics and metabolomics sciences are currently providing for cystic fibrosis (CF) research: from the understanding of cystic fibrosis transmembrane conductance regulator (CFTR) biology to biomarker discovery for CF diagnosis. Our work particularly focuses on CFTR post-translational modifications and their role in cellular trafficking as well as on studies that allowed the identification of CFTR molecular interactors. We also show how metabolomics is currently helping biomarker discovery in CF. The most recent advances in these fields are covered by this review, as well as some considerations on possible future scenarios for new applications.

## 1. Introduction

Cystic fibrosis (CF) is an autosomal recessive disorder that affects 1 out of 2500 newborns among Caucasians and it is caused by mutations in the gene encoding the cystic fibrosis transmembrane conductance regulator (CFTR) protein [1]. CFTR is a cAMP-regulated anion channel that transports Cl^−^ and HCO_3_^−^ across the epithelium in the respiratory tract, pancreas, gastrointestinal tract, the biliary and sweat ducts and part of the reproductive organs [2]. Cystic fibrosis is a multi-organ disease, with pathological changes occurring in organs that express CFTR, including secretory cells, sinuses, lungs, pancreas, liver and reproductive tract [1].

CFTR belongs to the ATP-binding cassette (ABC) transporter superfamily (ABC subfamily), and it is composed of five domains: two hydrophilic nucleotide-binding domains (NBD1 and NBD2) involved in ATP binding; two transmembrane domains (MSD1 and MSD2) with six hydrophobic α–helix segments each and a cytoplasmatic regulatory domain (R) with multiple phosphorylation sites involving cAMP-dependent phosphokinase A. Wild-type (wt)-CFTR presents two extracellular asparagine (Asn)-linked N-glycosylation sites between the transmembrane segments TM7 and TM8 [3]. The channel opening and closing are finely regulated by the alternative dimerization and dissociation of NBD1-NBD2: activation starts with phosphorylation of R domain through PKA and requires the binding of NBD domains to ATP and the consequent hydrolysis. Dephosphorylation of the R domain through protein phosphatases promotes the channel closing [4].

To date, over 1900 sequence variations in the CFTR gene have been reported [5], although, to date, pathogenicity has been demonstrated only for approximately 350 variants (CFTR2 database, https://cftr2.org). CFTR mutations have been divided into six classes, according to their final effects on CFTR: failure in protein synthesis due to frameshift or nonsense mutations (Class I); unstable conformation and early degradation of the protein (Class II); low transport of chloride through the channel (Class III); low ions conductance through the channel (Class IV); a small amount of functional protein (Class V) and destabilization at the apical membrane (Class VI) [6]. Nearly 90% of CF patients (with marked differences depending on ethnicity) show at least one copy of the most common mutation, a deletion of the phenylalanine in position 508 (F508del). The F508del-CFTR is incorrectly folded, accumulated in the endoplasmatic reticulum (ER) and subjected to proteasomal degradation; the small amount of protein that reaches the membrane is unstable and presents gating defects [7].

In the airways, chloride impermeability caused by mutated CFTR, and increased sodium absorption caused by the epithelial sodium channel, lead to a water/volume depleted periciliary liquid. Water loss increases mucus viscosity and impairs mucociliary clearance. Bacteria invading the cystic fibrosis lung are trapped in the viscous mucus layer on top of respiratory epithelial cells, leading to recurrent lung infections and chronic inflammation [8].

Although the first report on CF was published in 1938 by Dorothy Hansine Andersen [9], the CFTR gene was identified and cloned 50 years later, in 1989 [10]. Since its discovery, both the CFTR structure and function have been deeply investigated. These efforts improved our understanding of CF pathophysiology but also outlined the complexity of the disease. To improve our understanding of the role of CFTR, CF researchers now benefit from the most advanced techniques to investigate the biological systems at a global level: the omics. Omics represent a set of biomolecular disciplines, rapidly developed in the last 20 years, to investigate biological systems using large-scale molecular-level measurements, from DNA up to metabolic reactions. These techniques, together with the development of computational tools, allowed to couple high-throughput sample analysis with a global overview of the interactions established between different molecular levels within the cell. This paved the way for new investigation strategies and new therapies. Genomics was the first omics to be developed. It aims at characterizing the genome, in order to specify coding and non-coding sequences, their functions and structures. The project for the sequencing of the human genome, concluded in 2001, led to the identification of more than 20,000 coding genes [11]. While the DNA of a cell does not undergo marked changes during cell lifespan, protein and metabolite content usually varies over time, and depending on the tissue and physiological or pathological conditions.

The pathway that leads from DNA to protein synthesis and metabolites processing, involves numerous biochemical processes controlled at each step of the path. It is well known that metabolites play an essential role in gene regulation, an example being the role of cholesterol [12] or glucose [13]. Figure 1, Panel A represents how the genome is translated into a phenotype, passing through the corresponding proteo- and metabotypes [14], the left arrow represents the back process of regulation caused by the metabolites balance. The understanding of the connections between genomics variations and alterations in phenotype provides crucial insights into cell mechanisms, including the pathological ones.

While the CF-causing mutations on the CFTR gene are known, the resulting disease phenotypes at both metabolomic and proteomic levels are far from being fully elucidated. For these reasons, metabolic and proteomics studies were conducted to provide insight into the mechanisms by which CFTR mutations lead to disease effects.

Figure 1, Panel B represents a general workflow for untargeted proteomics and metabolomics experiments.

## 2. Proteomics

Proteomics aim to understand and characterize, both qualitatively and quantitatively, the protein content of a cell. The most broadly used technology in proteomics is LC–MS (mass spectrometry coupled to liquid chromatography) of protease-digested protein lysates (bottom-up proteomics, [15]) due to its unmatched ability to identify and quantify thousands of proteins in complex mixtures in a matter of minutes. Despite many advanced methods developed [16], the total coverage of the cell proteome has not been achieved yet. Many reasons lie behind this limit. When looking for molecular proofs of the translation of known coding genes, roughly 11% of the expected proteome is still missing [17]. In addition to this, as discussed above, the number of individual proteoforms exceed by orders of magnitude of the number of expected proteins.

## 3. Metabolomics

Metabolomics is the comprehensive study of the metabolites in cells, tissue and biofluids and provides an overview of the metabolic processes alterations resulting from the genetic background and environmental perturbations. The expression “metabolic profile” was used for the first time in 1971 [18], when advancements in technology allowed the shifting from qualitative to quantitative metabolomics. While proteomics deal with molecules that, in the end, are all chemically quite similar (peptides), metabolomics tracks an extremely broad and chemically diverse part of the cell chemical space, ranging from carbohydrates to lipids. This chemical diversity directly translates into the technology: while LC–MS represents the almost unique solution for proteomics, metabolomics is also frequently performed by GC–MS (gas chromatography coupled to mass spectrometry) and by NMR (nuclear magnetic resonance) spectroscopy. GC–MS, one of the oldest instrumental techniques, still offers unmatched performances for a relevant number of metabolomics applications [19], [20]. Despite being markedly less sensitive compared to MS, NMR is very powerful toward polar metabolites that are abundant in biofluids [21]; its high reproducibility makes NMR particularly suitable for clinical screenings [22] and the absolute quantification of the observed metabolites by NMR is an extremely important asset that MS currently does not offer. Proteomics and metabolomics data integration allows one to explore the interactions between metabolites and proteins: a crucial field of biology still largely unknown [23]. These and other activities go, with many others, under the so-called systems biology approach, a discipline of no clear-cut definition [24] that encompasses biology, mathematics and bioinformatics.

## 4. Proteomics to Investigate CFTR Network: Interactomics 

Interactomics is the study of protein interactions and the biological significance and function of them. Indeed, the interactors of a protein are the most likely candidates for the regulation of its activity, biosynthesis and proteolysis [25]. To have a comprehensive overview of how CFTR works in the context of a cellular network, studying CFTR interactome could reveal the pathways related to CFTR folding, trafficking and activity. Several studies have been conducted to explore wt-CFTR and mutant CFTR interactomes.

### 4.1. MudPIT-Based Interactomics

The first proteomics study to define CFTR protein interactors was conducted in 2006 [26]: Wang et al. applied the multidimensional protein identification technology (MudPIT) to study wild-type-CFTR interactome on lung and intestine cell lines (BHK, Calu-3, HT29 and T84). In this study, wild-type (wt) CFTR was immunoprecipitated and 167 of its interacting proteins were identified. In the same study, the interactome of F508del-CFTR was investigated by comparing the CFTR-specific proteomes immunoprecipitated from BHK cell lines expressing either wild-type or F508del-CFTR. This was the first study aiming at characterizing the differences in the interactome of wt vs. mutant CFTR. Indeed, since F508del-CFTR shows impaired folding and trafficking from ER to plasma membrane, the differential analysis of CFTR interactors at the ER level allowed the authors to highlight the pathways that are differently activated when the folded or the misfolded CFTR is transported. The authors found that both wt and mutant CFTR have strong interactions with a pool of chaperone and cochaperones proteins, highlighting the role of Hsp90 cochaperones that modulate Hsp90-dependent stability of CFTR protein folding in the ER. According to Wang and colleagues, failure of F508del to achieve an energetically favorable fold in response to the steady-state dynamics of the chaperone folding environment (the so-called chaperome) is responsible for the pathophysiology of CF [26]. In particular, the study revealed that the interaction of the Hsp90 cochaperone Aha1 is different for wt compared to F508del CFTR, and that silencing of Aha1 resulted in a rescue of F508del-CFTR activity at the plasma membrane. Interestingly, incubation of cells at 30 °C (a permissive temperature for CFTR maturation) provides a more energetically favorable folding environment leading to significant levels of cell-surface-localized F508del. Indeed, at reduced temperature, the F508del folding pathway appears to be more stabilized, and the folding properties of 37 °C and temperature-corrected F508del show different dynamics with respect to the Hsp90 cochaperone machinery. The most relevant conclusion of this work is that the chaperome can be adjusted to alter the dynamic relationship between those chaperone components specifically required for folding of CFTR, resulting in increased or decreased F508del mutant fold for export. Thus, the activity of cargo-associated chaperome components may be a common mechanism regulating folding for ER exit.

### 4.2. CoPIT-Based Interactomics

In 2015, another study identified a set of F508del-CFTR interactors [27], in particular those that potentially trigger the disease phenotype. To this purpose, a new immunoprecipitation technique, focusing on protein–protein interactions (co-purifying protein identification technology—CoPIT) was developed and applied to immortalized bronchial epithelial cell lines HBE14o-, expressing wt-CFTR and CFBE41o-, expressing F508del-CFTR. F508del-CFTR and wt-CFTR interactomes showed an overlap of more than 85% (368 proteins interact with both F508del- and wt-CFTR). Additional 208 and 62 proteins were observed interacting exclusively with F508del-CFTR and wt-CFTR respectively. Among the specific interactors of F508del-CFTR, they found chaperones known to be involved in ER quality control (such as Hsp90) as well as proteins already reported to be involved in CFTR degradation, like AMFR, STUB1 (CHIP) and VCP. Interestingly, the study also highlighted a set of proteins (including AUP1, SEL1L and FAF2) already known to be involved in ERAD (ER-associated degradation pathway) of misfolded proteins but never before associated to CF. CoPIT was also applied to investigate changes in the interactome of F508del-CFTR following different rescue maneuvers, including cell incubation at a permissive temperature for CFTR processing (26–30 °C) and pharmacological treatment with histone deacetylase (HDAC) inhibitors that, by modulating proteostasis [28], can lead to an increase in stabilization, trafficking and activity of F508del-CFTR channel. Consistently with previous data derived from transcriptomic studies [29], the temperature shift extensively changed the F508del-CFTR interactome (more than 65% of protein interactors were remodeled). The altered proteins in the interactomes can be divided into three classes: proteins involved in degradation (belonging to ubiquitin-mediated pathways, ERAD system and in endocytic removal of plasma membrane proteins); proteins involved in folding processing (such as Hsp90 and glucose-regulated proteins); proteins involved in RNA processing (such as PABPC1, involved in mRNA metabolism). On the contrary, treatment with HDACi abolished interaction highly specific for F508del-CFTR, restoring some interactions peculiar of wt-CFTR.

### 4.3. Elucidating the AFT-Based CFTR Interactome

In 2018, a study from the group of Carlos Farinha [30] focused on understanding the regulation of F508del-CFTR retention in the ER, controlled by ERQC (endoplasmic reticulum quality control) machinery. They used MS-based proteomics and bioinformatics tools to identify proteins that specifically interact with CFTR in motifs important for ER checkpoints. In particular, they focused their interest in the AFT (arginine-framed tripeptide) motif, used by the cell as a negative signal to retain unfolded proteins in the ER [31], and in the di-acidic motif, responsible for the association with COPII machinery for the ER export [32]. The folding status of CFTR impacts the exposure of these motifs, responsible either for CFTR retaining in the ER (and subsequent degradation through the proteasome machinery) or for CFTR exit from ER. Farinha and collaborators reported that export-incompetent CFTR variant, i.e., in which both aspartate residues were changed to alanine (D565A and D567A, termed DD/AA-CFTR) showed a decreased association with proteins in vesicle-mediated transport and proteins involved in trafficking pathways, such as GET4 and TRIP10, known to be involved in the degradation of misfolded proteins in ER [33]. Among the proteins strongly interacting with CFTR, the authors identified general folding/proteostasis components, such as GEFs for trafficking, GTPases such as CDC42BPG or ARHGEF1 or sensors such as UGGT1, supporting the idea that the recognition of AFT motifs is a general mechanism for folding assessment relying on ubiquitous components of the cellular machinery. Interestingly, the list of protein interactors of AFT motifs also includes proteins previously identified, using a functional genomics approach, as possible regulators of F508-CFTR fate, like UBE2I, UBA52, UBA2 or CHD4 [34]. This study was the first introducing the concept that CFTR interactome is much more shaped by the folding status of the protein (as assessed by its ability to exit the ER and reach the plasma membrane) than by its subcellular location. All the previous studies were performed to explore the protein networks of F508del-CFTR.

### 4.4. CFTR Interactomics at Plasma Membrane

In 2019, Matos et al. [35] provided the first study of CFTR interactome at the plasma membrane. By using an engineered immortalized bronchial cell line (mCherry–FLAG–CFTR CFBE cells [36]), they developed an immunoprecipitation-based protocol, in the presence of a cross-linking agent, capable of selectively capturing the CFTR interactome at the plasma membrane. They then studied the interactomes of both wt and F508del-CFTR upon pharmacological treatment to find differences in protein interactors at the plasma membrane. Their study highlighted 22 proteins involved in macromolecular protein complexes localized either in the plasma membrane or in the cytoplasm. Among them, three proteins were selected as relevant interactors belonging to the NHERF1-ezrin complexes: 14-3-3 zeta, calpain-1 and importin 5, with calpain-1 being the most promising one, given its reported druggability. Indeed, calpain-1 is a calcium-sensitive cysteine protease whose inhibition was reported to promote F508del-CFTR rescue in peripheral blood mononuclear cells (PBMCs) in CF patients [37]. Their findings unveiled the role of calpain-1 as an exclusive F508del interactor that prevents active ezrin recruitment, impairs F508del anchoring to actin and reduces its stability in the plasma membrane. More importantly, their study also confirmed that calpain 1 down-regulation or its chemical inhibition dramatically improves the functional rescue of Phe508del-CFTR in airway cells.

Figure 2 summarizes the main studies reported in this review and the corresponding major findings.

### 4.5. G551D-CFTR Interactomics

To the best of our knowledge, only one study was conducted to study the interactome of other CFTR mutations: the interactomic study, by Teng et al. [38], performed on G551D-CFTR, a mutant channel with normal expression on the cell surface, but impaired channel activation, associated with severe disease due to its altered channel activation. The aim of the study was to identify specific interacting proteins of G551D-CFTR, based on the idea that pharmacological rescue of this mutant might occur through a direct or indirect binding, stabilizing an intramolecular interaction [38]. The authors, by using HeLa cells transfected either with wt-CFTR, as a control group, or with G551D-CFTR, used 2DGE (bidimensional gel electrophoresis) followed by MS protein identification to generate the hypothesis that calumenin plays a role in the physiopathology of G551D-CFTR.

## 5. Analysis of CFTR Post-Translational Modifications 

Proteomics is also widely used to study post-translational modifications (PTMs), known to be extremely relevant for protein signaling and activity. The investigation of PTMs of the mutant proteins in comparison to the wt helps to understand the mutant processing and could enable the development of new therapeutic strategies. In 2019, Pankow et al. investigated how some PTMs [39] (such as phosphorylation and methylation) regulate the turnover of CFTR, studying the differences in PTMs landscape caused by mutations of CFTR and whether these changes have a role in CFTR rescue upon treatment. It was already demonstrated that misfolding of F508-CFTR affects the PTMs-regulated activity of the channel [40,41] but this is the only study in which PTMs were reported as key regulators of CFTR trafficking and biogenesis. As a first step, the authors developed a method to study PTMs in both 16HBE (expressing wt-CFTR) and CFBE41o- (expressing F508del-CFTR), which led to the mapping of 80% of CFTR sequence and the identifying of 37 PTMs sites (20 phosphorylations, 6 methylations and 11 ubiquitinations). Some of these PTMs were never reported before, like methylation sites, prevailing in wt-CFTR and ubiquitination sites, mostly detected in F508del-CFTR. They then quantified specific PTMs, revealing a reduction in the methylation of three amino acids (Lys442, Lys584 and Arg751) and a reduction of phosphorylation at the regulatory insertion (RI) element (Thr421, Ser422 and Ser427) of F508del-CFTR in comparison to wt-CFTR [39]. In addition, the authors quantified the phosphorylation abundance in the RI element in four CFTR mutants: G551D, R117H, N1303K and F508del. Their studies showed that the amount of phosphorylation in G551D and R117H mutants (having a defective ion channel activity with no effects on biogenesis) was comparable to the amount of phosphorylation in wt-CFTR. On the contrary, the phosphorylation at Thr421 to Ser427 was drastically reduced for N1303K- and F508del-CFTR (both having a defective maturation), suggesting that these phosphorylation sites are crucial for CFTR biogenesis. Figure 3 shows the structure of CFTR along with the reported PTMs and most frequent mutations.

To better understand the role of PTMs in the CFTR maturation process, Pankow and colleagues [39] analyzed the changes of PTMs of CFTR upon several rescue strategies, such as treatment with different compounds or the abolition of CK2α, a kinase expressed in the cytoplasmic side of the ER and Golgi apparatus known to have a role in the phosphorylation of RI element sites of CFTR [42]. Based on the experimental evidence collected, Pankow and colleagues were able to demonstrate that wild-type CFTR hotspots (sequences of roughly 50 amino acids in length, in which multiple PTMs cluster together) were phosphorylated and contained methylation sites at the N or C terminus of each hotspot [39]. However, the same sites in F508del CFTR hotspots were preferentially ubiquitinated instead of methylated, and the F508del CFTR hotspots contained altered phosphorylation amounts as well as additional phosphorylation and ubiquitination sites [39]. Moreover, they demonstrated that the rescue of F508del-CFTR is tightly associated to the increase in the amount of phosphorylation at RI element sites (Thr421, Ser422 and Ser427), together with a reduction of ubiquitination of Lys420 and methylation of Lys442 [39]. This pattern of mutually exclusive methylation (wild type) and ubiquitylation (F508del) points to a more elaborate PTM code in which phosphorylation sites are preceded or followed by lysines that can be differentially modified to regulate specific functions or steps in CFTR biogenesis [39]. To be noted that Pankow et al. [39] did not limit the PTMs investigation to F508del exclusively, but also explored the PTMs status of other CFTR mutants. This study is particularly relevant, because it is the first underlining the essential role of PTMs in protein folding and quality control during CFTR maturation. It will be interesting to study the differences of the interactomes in the absence of key PTMs on CFTR protein and how PTMs modulate CFTR interactome. In addition, it will be also interesting to explore whether such a PTM code exists in other membrane proteins to regulate biogenesis.

## 6. Global Changes in Protein Expression Observed in CF

Global expression proteomics seeks the simultaneous identification and quantification of as many proteins as possible in a cell. Currently state-of-the art technology enables the reliable quantification of several thousand proteins in a single analysis [43]. In comparative expression proteomics, changes in protein expression levels in a given cell, tissue or organism are measured under different conditions or challenges. Since the proteins are the ultimate effectors of most cellular processes, the differentially expressed proteins could uncloak the pathophysiology of the disease or could be part of the body’s response to the disease. Only a few studies were conducted to generate a protein expressional profile in CF bronchial tissue. In 2006, Frischer et al. performed a global protein expression pattern analyzing the bronchial biopsies of nine CF patients in comparison to those of eight control subjects [44]. This comparison was conducted using 2DGE and subsequent protein identification through mass spectrometry. More than 300 differently regulated proteins were identified, with a number of them, involved in inflammation, infection and the cellular stress response, differentially expressed in bronchial tissues of CF vs. non-CF individuals. In 2018, using a similar approach on CFBE41o- and 16HBE14o- cell lines, Puglia et al. identified 15 proteins significantly dysregulated by the F508del mutation [45]. A limitation of many studies on global protein expression is that they rely on 2D-GE, which, besides suffering from a limited dynamic range, it is biased toward the most abundant and soluble proteins. The first non-gel, shotgun, label-free quantification of total lysates was performed by the Yates’ group in 2014 [46] analyzing the CFBE41o- proteome in comparison to that of 16HBE14o- cell lines. The use of this technique allowed them to deeply investigate this proteome, obtaining information about thousands of proteins possibly important for either understanding the molecular mechanism of the disease or as novel molecular markers. Yates and collaborators were able to identify 349 differentially expressed proteins (218 up- and 131 down-regulated), some of them belonging to biological processes directly linked to CF, such as ubiquitin-mediated degradation, response to unfolded protein, regulation of proteolysis, protein folding and endocytosis [46]. A further step towards understanding the impact of CFTR mutations on bronchial cell proteome was made possible by using primary human bronchial epithelial cells that directly reflect the CF pathophysiology, including the heterogeneity of individual patients. By using SWATH (sequential window acquisition of all theoretical mass spectra) label-free proteomics, Braccia et al. [47] identified 154 proteins dysregulated by the CF pathology (94 upregulated and 60 downregulated), including known CFTR interactors (such as CHIP) as well as proteins not previously known to be related with CF. We analyzed the contribution of proteomics to basic CF research and its application to understand and characterize CFTR trafficking.

## 7. Metabolomics and CFTR Research

The contribution of metabolomics in investigating basic CF pathophysiology is, so far, quite limited. Some excellent papers report on the role of lipid composition on CFTR trafficking: Bear’s group, for example, [48,49] extensively investigated the role of cholesterol and some sphingolipids on stability and regulation of CFTR. Metabolomics, on the other hand, plays a prominent role in the identification of biomarkers associated with CF in human subjects. Wetmore, et al. [50] conducted the first untargeted metabolomic study in 2010. With the aim of understanding the epithelial dysfunction caused by CF, they performed three independent metabolomic analysis comparing CF patients, homozygous for F508del mutation, to non-CF subjects using primary human airway epithelial cells. From this study, around 400 metabolites were detected and more than 100 positively identified. The statistical and KEGG (Kyoto Encyclopedia of Genes and Genomes) pathway analysis revealed a list of altered processes, like:Nucleotide metabolism, with a decrease in purine biosynthesis, essential for the control of ASL (airway surface liquid) volume. The authors claimed that these results are in contrast with previous results from EBC (exhaled breath condensate) purinergic receptors [51].Increase in the catabolism of tryptophan, causing accumulation of molecules associated with oxidative stress [52].Reduction of glutathione biosynthesis, a key regulator of the oxidative status of the cells.Decrease in osmolytes, such as sorbitol and glycerophosphorylcoline, which have a role in the regulation of cell volume.Low levels of glucose metabolism, a possible cause of an increase in cell sensitivity to oxidative stress.

Interestingly, this study highlighted some metabolic alterations that were never associated to CF before, thus providing new insights into the understanding of the disease. In 2014, Joseloff et al. [53] performed an extensive metabolomic study of serum of CF children. This was the first study investigating the serum of 31 children affected by CF to detect differences in comparison to the serum of 31 non-CF children. Of 289 metabolites identified (out of 459 features detected), 92 were significantly altered by CF. A KEGG pathway analysis of the altered metabolites identified novel pathways involved in cellular energy production through the β-oxidation of fatty acid, as well as previously reported ones, such as increased oxidative stress. Very recently, an untargeted metabolomics study on 39 adult CF patients demonstrated that sorbitol is greatly decreased in these subjects [54]. Alterations in alditols biosynthesis were already observed in cell models [50] and might contribute to explain why patients benefit from alditols inhalation [55].

Untargeted metabolomic analysis has been applied also to collect data on the efficiency of drug treatments, especially for those molecules whose clinical effectiveness is still under evaluation. In 2018, Kopp et al. [56] performed a metabolomic study to find systematic changes caused by the Lumacaftor/Ivacaftor treatment that shows variable clinical responses among CF patients homozygous for F508del-CFTR. The authors analyzed the serum of 20 CF patients pre- and post-six-month therapy. Despite the relatively low number of samples, they identified a set of metabolic pathways altered by the treatment. Among the top 30 dysregulated metabolites, bacteria-associated metabolites involved in lipid and amino acid metabolisms were found. Moreover, the analysis detected an increase in bile acid levels and a decrease in specific lipids, phospholipids, sphingolipids and ceramides, whose role of ceramides in CF has been already reported [57]. Bile acids have also been used by our group to help the evaluation of effective CFTR rescue in a mouse model of CF [58]. The abolition of RNF5 (a known CFTR interactor) decreases to almost the wt condition levels of bile acids in mice stool, by improving their intestinal reabsorption.

## 8. Metabolomics for Biomarker Discovery: Support and Aid CF Diagnosis

Metabolomics has been recently explored as a valuable tool to support CF diagnosis. The protocol of neonatal screening is based on the determination of immunoreactive trypsinogen (IRT), a pancreatic enzyme highly secreted into the bloodstream in CF patients because of pancreatic ducts blockages. To evaluate the levels of IRT, a drop of blood is collected on absorbent paper (dried blood spots—DBS) on the third day of life. Although newborn screening protocols positively impacted the development of new diagnostic methods, they have several limitations related to the low specificity and the complexity of consensus guidelines [59]. Thus, the gold standard diagnosis remains the pilocarpine-stimulated iontophoresis sweat test (which is, in se, a metabolomic approach) [60]. The concentration of chloride in the sweat above 60 meq/L is a diagnostic criterium used from the first days of life, to be eventually confirmed by the presence of mutations in the CFTR gene. In addition to this, tests of functionality of the protein may be carried out, such as the measurement of the potential difference at the level of the nasal mucosa (nasal potential difference) [61] or the measurement of the intestinal current after a biopsy [62]. Several studies on different biofluids aim at identifying biomarkers able to come across the limits of normal screening protocols as well as discriminating between a highly heterogeneous genotype of CF cases.

The first untargeted metabolomics study for the identification of presymptomatic CF newborns was carried out using HRMS on dried blood spots [63]. Authors aimed firstly to differentiate between asymptomatic CF newborns and healthy ones then to distinguish true CF from screen-positive (SP)/non-CF newborns. They found 32 metabolites differently expressed in CF newborns, in particular several circulating amino acids significantly decreased in CF compared to screen positive/non-CF neonates leading to the conclusion that CF newborns suffer from protein maldigestion/malabsorption, resulting from exocrine pancreatic insufficiency [63]. Interestingly, N-glycated amino acids are among the altered metabolites in CF newborn. These molecules are markers of hyperglycemia caused by non-enzymatic glycation of free amino acids, commonly associated with chronic diseases [64]. In the same study, low levels of circulating glutathione in CF newborns were found, suggesting that the nucleotide-regulated glutathione efflux in the exocrine epithelial cells is impaired in CF. The reduction of glutathione was already described [65] and it contributes to oxidative stress and reduced antimucolytic activity [66].

A comprehensive characterization of sweat from screen-positive CF infants was carried out in a recent study [67], where sweat samples from CF affected infants were compared with CF unaffected individuals by multisegment injection−capillary electrophoresis−mass spectrometry (MSI–CE–MS). Many dysregulated metabolites were detected and associated with the CF disease status. In particular pilocarpic acid (PA, a hydrolysis byproduct from pilocarpine used for sweat testing), L-asparagine (Asn), mono (2-ethylhexyl) phthalic acid (MEHP, a metabolite found in many biofluids and derived from environmental exposure to bis(2-ethylhexyl) phthalate, DEHP) and glutamine (Gln). Surprisingly the two exogenous metabolites PA and MEHP, strongly correlated to pilocarpine-stimulated sweat secretion were found to be lower in CF infants with respect to unaffected screen-positive controls. PA is metabolized by the enzyme human paraoxonase 1 (PON 1) [68]; the enzyme family is also involved in modifying prenatale phtalate exposures [69]. These results suggested a correlation between the decreased levels of the two exogenous metabolites and a possible impaired activity of paraoxonase in CF individuals.

In order to obtain a non-invasive sampling method and to reduce time and variation in sample preparation steps, a recent lipidomic study aimed to find an alternative way for CF diagnosis by combining desorption electrospray ionization mass spectrometry and a machine-learning algorithm based on gradient boosted decision tree. With this method, the authors recognize patterns that distinguish CF from non-CF patients [70] in perspiration samples.

Lipidomic is also important to explore CFTR environment at the plasma membrane. While lipids are known to affect the function and the stability of other ABC transporters [71], the relation between lipid composition and CFTR is still extensively investigated. Many lipids are known to be imbalanced in CF [72]:Free fatty acids, like linoleic [73] and arachidonic acid [74], are decreased and increased in nasal epithelial cells collected from CF patients respectively.Cholesterol, regulator of plasma membrane fluidity, is increased in cells [75] and tissues [76] expressing mutant-CFTR (several mutations were studied, including F508del and G551D).Ceramides, known to be involved in proinflammatory and proapoptotic signaling, were thought to be increased in CF patients [77]. Based on these studies, amitriptyline and fenretinide, two drugs modulating ceramide metabolism, are undergoing clinical trials. Nevertheless, recent studies showed that the plasma levels of ceramides in CF murine models are decreased [78]; these unclear results underline the need of further investigations of the role of sphingolipids in CF. Indeed, sphingosines levels are lower in CF mouse epithelial cells compared to no-CF epithelia [79] and gangliosides are reduced in Calu-3 bronchial epithelial cells (found to be decreased) [80].

All these evidences point at a very significant role of lipids in CF pathology.

## 9. Omics to Study the CF Microbiome

The composition of the lung microbiome in CF patients has been widely investigated as it changes during lung damage progression as well as during antibiotic therapies [81].

In addition to the lungs, the digestive system is also seriously impaired in CF patients. In many CF individuals (roughly 80%), the pancreatic ducts are obstructed with sticky mucus, which leads to an inflammatory response and tissue destruction. The decreased secretion of digestive enzymes causes chronic pancreatitis and even pancreatic insufficiency, which can lead to malnutrition and hence retardation in growth. Problems with digestion can lead to diarrhea, malnutrition, poor growth and weight loss. In adolescence or adulthood, a shortage of insulin can cause a form of diabetes known as cystic fibrosis-related diabetes mellitus. Many studies showed that the gut microbiota enterophenotype of patients with FC is an expression of alterations observed at the intestinal level [82]. Recent NMR and HRMS metabolomics studies carried out on fecal samples from pediatric and adult subjects [83] show that the gut microbiota composition of CF subjects differs significantly from healthy non-CF controls. In pediatric patients, the intestinal manifestation of the disease is represented both by the gut CF enterophenotype and by the production of bacterial CF biomarkers. These results suggest the importance of devising therapies aiming at counteracting the malnutrition through microbiome analysis in pediatric patients. In order to demonstrate the potential of fecal metabolomics in CF, a recent survey [84] compared CF pancreatic insufficient (PI) and sufficient (PS) children with healthy controls (HC). They compared stool metabolites between: PI-CF with and without evidence of intestinal inflammation (and on pancreatic enzyme replacement therapy—PERT), PS-CF and healthy controls (HC). The principal component analysis highlighted a clear difference between PI-CF versus PS-CF and HC; moreover, the level of a non-identified feature increased with the severity of disease (HC versus PS versus PI); among features related to the inflammation they identified lipoyl-GMP as a potential inflammatory biomarker and a high level of glycerol 1,2-didodecanoate 3-tetradecanoate in PI-CF with inflammation.

The impairment of the respiratory system represents the most frequent clinical manifestation of CF. It starts early in life, degenerates rapidly and it is the main cause of mortality. The CF condition leads to severe and frequent infections with bronchiolitis and hemoptysis; the main microorganism that colonizes the airways in CF is *Pseudomonas aeruginosa*, although other bacterial species are frequently detected, such *Propionibacterium*, *Staphylococcus aureus* and Clostridiaceae, including *Clostridium difficile*, and a low abundance of *Eggerthella*, *Eubacterium* and *Ruminococcus*. Bacterial colonization results in acute and chronic infections, recurrent pulmonary exacerbations (PEx) and irreversible loss in lung function, with substantial morbidity and mortality [85]. The characterization of the bacterial charge is of primary importance in the therapy against CF. Therefore, the research of biomarkers of inflammation, infection and tissue degeneration has a fundamental role in the prediction of the disease manifestation, allowing to deal with the disease in the early stages of its progression. Hundreds of studies investigating inflammation were carried out, in order to highlight biomarkers of infection. Microbial metabolites may predict the onset of PEx in a proper time for therapeutic intervention or prevention and monitoring lung disease [86,87]. Many circulating blood-based biomarkers, such as a C-reactive protein (CRP), were widely studied [88] with different techniques. A pilot study in 2019 [89] used multiple reaction monitoring mass spectrometry (MRM–MS) to evaluate the use of a panel of blood proteins to predict short-term PEx risk in children and adolescent. The proteins (C-reactive protein, CRP; peroxiredoxin-2, PRDX2; hemoglobin subunit alpha, HBA; carbonic anhydrase 1, CAH1; cluster of differentiation 5, CD5 and apolipoprotein C-II, APOC2) were selected on the base of a previous study [90]. They collected plasma samples from a characterized cohort of placebo arm CF-patients involved in a clinical trial evaluating azithromycin. The panel of proteins was monitored during scheduled visits and compared with clinical factors, such as ppFEV1, and candidate blood biomarkers like absolute neutrophil count. The study is a proof-of-concept for monitoring inflammation parameters in order to predict PEx.

The metabolomics investigation of the CF phenotype has also focused on identifying specific biomarkers in body fluids other than blood. Volatile metabolites have been extensively investigated in the quest for new CF biomarkers that could be reliably associated with lung inflammation. Metabolomics of exhaled breath condensate (EBC) [91,92] proved to be a fast and non-invasive method to identify biomarkers related to inflammation. Very interestingly, a number of studies on EBC pointed out at HCN as an important biomarker in *Pseudomonas* cultures [93,94,95,96], along with many others metabolites, like hydrocarbons and N-methyl-2-methylpropylamine [97,98]. Sputum has also been extensively investigated as a potential source of biomarkers [99,100].

## 10. Conclusions and Future Perspectives

In this review, we summarized some of the fields of applications of metabolomics and proteomics to CF research. We demonstrated that these two omics sciences already provided extremely useful contributions to the understanding of the molecular mechanisms underlying the pathology of CF and to the identification of molecular biomarkers. We underlined the role of proteomics in pinpointing the role of PTMs in the regulatory mechanism of CFTR trafficking and regulation, with tens of PTMs site identified and characterized. We also reviewed how proteomics allowed the identification of specific CFTR interactors, like Calpain-1 and Aha1, which play a major role in CFTR rescue. Quite interestingly, beside CFTR, proteomics has also been used to investigate other ion channels, from both the structural and functional standpoints [101,102] or to elucidate mechanisms behind protein–protein interactions between receptors and regulatory partners [103]. Lipidomics has also become an important player in understanding how lipids act on the stabilization of membrane channels [104,105].

We also showed that metabolomics also has the potential to significantly contribute to the investigation of basic CF mechanisms. Indeed, we believe that a large-scale systematic investigation of the alterations on cell metabolome (and lipidome) induced by CF in in vitro models is still missing.

We then reviewed the very significant role of metabolomics in CF biomarker discovery as well as its contribution to the elucidation of gut microbiota alterations.

From this overview, it appears clear that other fields of applications to CF research can be envisaged. We hereby suggest some ideas for future investigations. CFTR interactome has been extensively investigated and CF research now has a somewhat clear picture of the proteins that act on CFTR trafficking. Quite surprisingly, though, we could not find in the literature an extensive investigation of CFTR interactome done by crosslinking studies. This technique has now become a very powerful tool to investigate protein trafficking [106], with new crosslinking agents able to work on intact cells. The corresponding field of mass spectrometry is currently achieving breakthrough results, due to development of MS-cleavable crosslinkers [107] and dedicated software tools able to automatically scan thousands of MS/MS spectra of peptides for crosslinked structures [108]. Proteome-wide investigation of protein–protein interactions at their naïve state are thus ready to be routinely applied to CF research. While collecting material for this review, we also realized that, in the applications of omics to basic CF research, large scale, multiomics studies are still missing. Indeed, being able to efficiently integrate data from different “omes” (genome, proteome and metabolome) together in a global overview of the chemical space of a CF cell would definitely be a breakthrough advancement. Very few studies focused on CFTR mutations other than F508del. This is of course not surprising, since the worldwide incidence of F508del mutation is approximately 75-80% of the alleles, while the other “frequent” CFTR mutations hardly accounts for 1% as overall frequency [109]. Finally, while preparing this article, we also noted that not all the omics papers we read clearly report to have uploaded the corresponding RAW data on publicly available data sharing servers, like PRIDE [110] or METABOLIGHTS [111]. This is a very important point of discussion in both the proteomics and metabolomics communities. Only a part of the journals, and over the last few years only, made the public data sharing mandatory for publishing. It is crucial for the whole CF community that all data, and particularly those collected from human subjects, are made available and shared in their native RAW form. Many tools have been developed to “translate” data (particular mass spec ones) in platform-independent formats and it is now relatively easy to extract biological data (protein identity and abundance, for example) from virtually any mass spec experiment ever performed. If we consider the power of current artificial intelligence and machine learning algorithms in extracting meaningful results from big omics data [112], we realize why, under many aspects, the contributions of omics to CF (and biomedical research in general) is, in the end, still in its infancy.

## Figures and Tables

**Figure 1 ijms-21-05439-f001:**
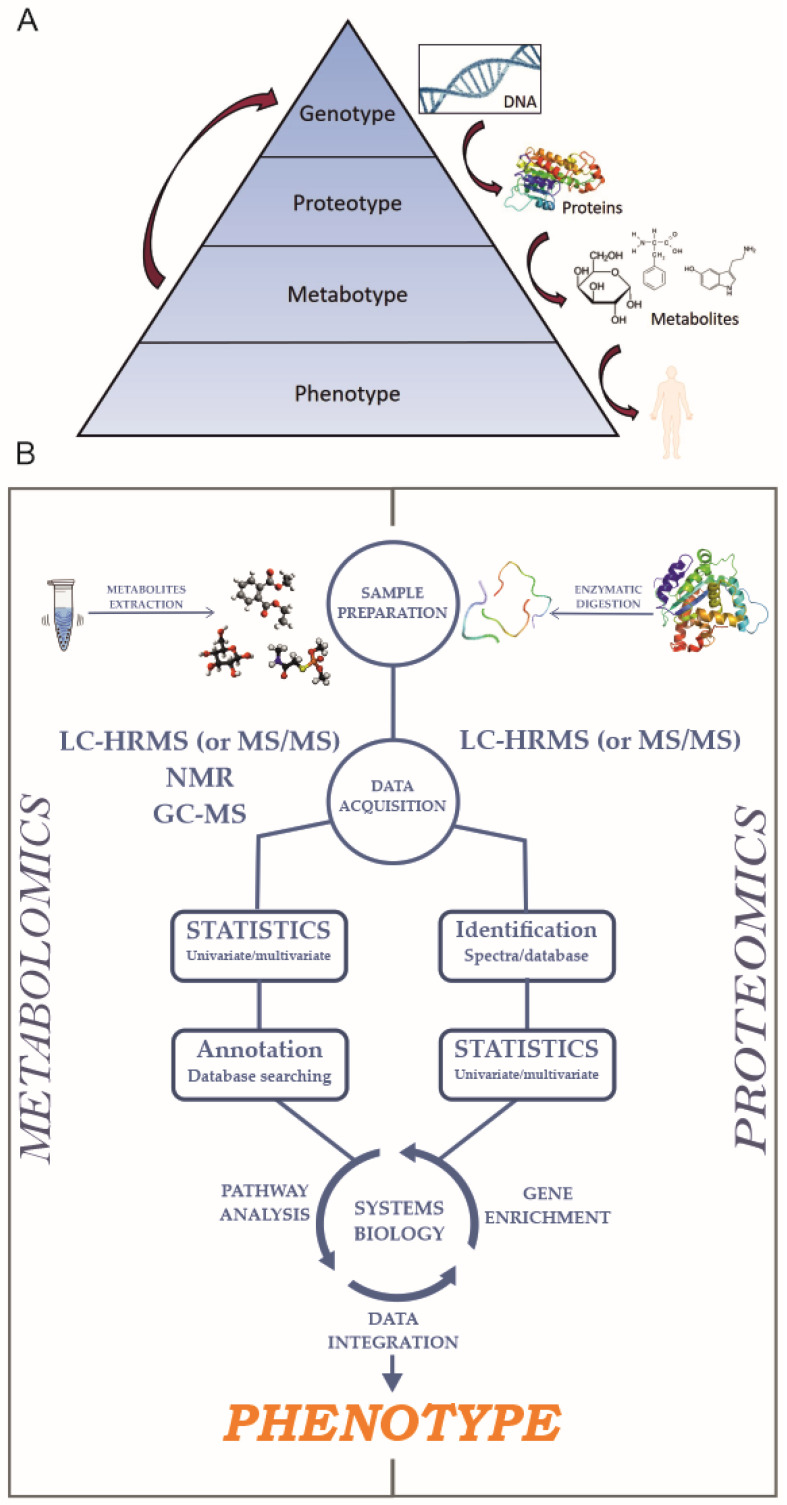
(**A**) Cartoon showing the pathway from the genotype to the phenotype. (**B**) Schematic representation of untargeted proteomics and metabolomics workflows.

**Figure 2 ijms-21-05439-f002:**
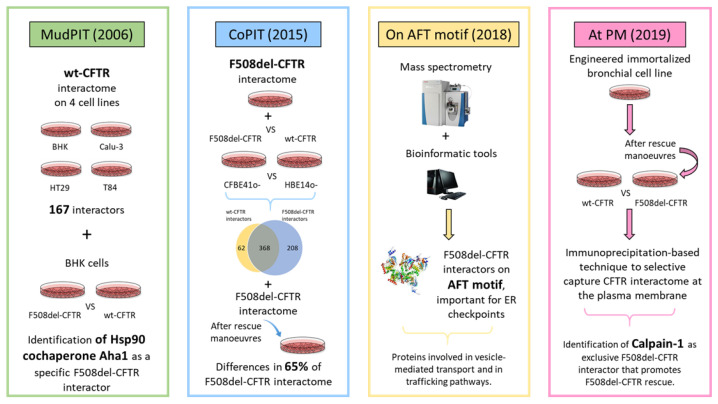
Summary and outcomes of four of the cystic fibrosis transmembrane conductance regulator (CFTR) interactomic experiments highlighted in this review.

**Figure 3 ijms-21-05439-f003:**
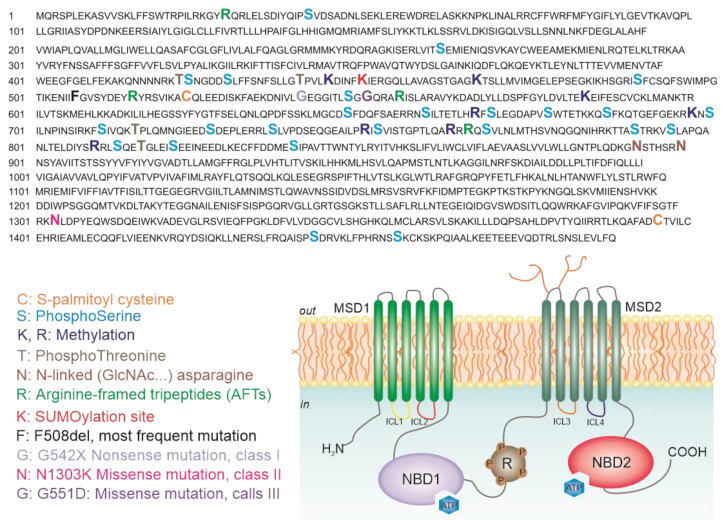
Representation of CFTR structure at the plasma membrane with known CFTR PTMs indicated in the primary sequence.

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
