# Peer review of "Proteomics and Metabolomics for Cystic Fibrosis Research"

_ijms, 2020, doi:10.3390/ijms21155439_

Round 1
Reviewer 1 Report
The authors present an adequate review of NMR and MS based research in cystic fibrosis. Roughly 1/3 of the paper and the only graphic are dedicated to going over the different analytical strategies. While this may be useful for readers without an analytical chemistry background, there are numerous reviews on the topic and this section could be trimmed down. The focus of the review should be amalgamation of the knowledge gleamed from these techniques about CF. Some figures on CF and CFTR would be very helpful in presenting this information. For example, a diagram documenting known PTMs and mutations of CFTR, or another diagram or table presenting the known interactors would increase the value of this review.
Additional points:
Line 110 in MS absolute quantitation is achieved through stable isotope dilution.
Line 123 in high resolution MS some sensitivity is lost because mass resolution and accuracy is increased (doesn't have anything to do with scan range).
Proteoform is mentioned but there is no discussion about top-down proteomics, perhaps because it's not yet been applied to CF?
Section 4 hops between discussion of metabolomics and proteomics which is distracting. It would be better to separate the two topics.
Author Response
The authors present an adequate review of NMR and MS based research in cystic fibrosis. Roughly 1/3 of the paper and the only graphic are dedicated to going over the different analytical strategies. While this may be useful for readers without an analytical chemistry background, there are numerous reviews on the topic and this section could be trimmed down. The focus of the review should be amalgamation of the knowledge gleamed from these techniques about CF. Some figures on CF and CFTR would be very helpful in presenting this information. For example, a diagram documenting known PTMs and mutations of CFTR, or another diagram or table presenting the known interactors would increase the value of this review.
Dear reviewer,
many thanks for your constructive comments. After receiving your suggestion about the introduction, we extensively changed this part. As per your suggestion, we have also added two additional figures, showing CFTR PTMs and structure and a graphic recap of the major investigations of CFTR interactors.
Additional points:
Line 110 in MS absolute quantitation is achieved through stable isotope dilution.
Line 123 in high resolution MS some sensitivity is lost because mass resolution and accuracy is increased (doesn't have anything to do with scan range).
Proteoform is mentioned but there is no discussion about top-down proteomics, perhaps because it's not yet been applied to CF?
Section 4 hops between discussion of metabolomics and proteomics which is distracting. It would be better to separate the two topics.
Thanks. Please see our revised version of the manuscript.
Reviewer 2 Report
In their review “Proteomics and Metabolomics for Cystic Fibrosis Research” the authors describe the application of proteomics and metabolomics to the study of cystic fibrosis. Their coverage of the field seems to be quite extensive. However the most interesting part of the review discusses the studies on the interactome of CFTR, a field that could be considered separate from proteomics; the authors should consider adding “interactomics” to the title and describe it in the introduction.
Specific issues:
- The introduction discusses various omics fields, mentions transcription (line 55), but does not mention transcriptomics.
- Figure 1B is oversimplified and not up to date, often a metabolomics also starts with identification and (relative) quantification of metabolites, before statistical analysis. The field moves more and more in that direction.
- Line 65-67: metabolomics is still mostly applied to biofluids, so mentioning changes on the cell metabolome caused by dietary changes, increased physical activities etc. needs references.
- The section on interactomics switches between the interactome of CFTR in the ER, to the interactome of CFTR while in the plasma membrane (line 206), back to the interactome in the ER (line 221), which is confusing. I suggest rearranging this part.
- Line 313: “we analyzed the contribution of proteomics...” while the next line suddenly starts discussing metabolomics. This seems to be in the wrong section.
Author Response
Dear reviewer,
Thank you for your comments to our review. Following your suggestions, we agree with you about the interactomics paragraph. We re-organized it into sub-paragraphs and we added a figure summarizing all the interactomics experiments cited by the review. We prefer not to change the title.
Specific issues:
- The introduction discusses various omics fields, mentions transcription (line 55), but does not mention transcriptomics.
- Figure 1B is oversimplified and not up to date, often a metabolomics also starts with identification and (relative) quantification of metabolites, before statistical analysis. The field moves more and more in that direction.
We agree with this point, but this refers to targeted metabolomics then. Figure 1 was meant to depict a general untargeted workflow. This is now indicated. Thanks for this point.
- Line 65-67: metabolomics is still mostly applied to biofluids, so mentioning changes on the cell metabolome caused by dietary changes, increased physical activities etc. needs references.
This is exactly one of the topics that we believe might be addressed by future research activities. We are indeed citing papers (from Bear’s group, for example) that report on the role of lipids and metabolites in the basic understanding of CFTR trafficking and stability.
- The section on interactomics switches between the interactome of CFTR in the ER, to the interactome of CFTR while in the plasma membrane (line 206), back to the interactome in the ER (line 221), which is confusing. I suggest rearranging this part.
- Line 313: “we analyzed the contribution of proteomics...” while the next line suddenly starts discussing metabolomics. This seems to be in the wrong section.
Thanks, as indicated above, these parts have been extensively rearranged.
Reviewer 3 Report
General impression:
The review would be more appealing if more focused on what information on CFTR/CF could be gained from proteomic/metabolomic studies. Specifically, which pressing questions can potentially be answered using proteomics/metabolomics? Lengthy introductions about the various OMICs approaches should be removed. CFTR needs a more extensive introduction. The successful use of proteomics and metabolomics to study other ABC proteins and other ion channels should be discussed, to show how the CFTR field can benefit from these OMICs approaches.
Specific comments:
- Unnecessary lengthy introductions are given for genomics, proteomics and metabolomics. Extensive descriptions of sample workup/processing, statistical analysis etc. can be removed. All readers interested in CFTR will know that genes encode proteins and that large parts of the genome consist of non-protein coding regions.
- CFTR deserves a more thorough introduction and should include the fact that CFTR belongs to the ATP-binding cassette proteins and is encoded by ABCC7. The general structure of CTFR should be given. e.g. ATP hydrolyzing domains involved in gating of the channel, the regulatory R domain that can be phosphorylated regulating its open probability etc. Moreover, when discussing the applicability of proteomics and metabolomics to study CFTR, other ABC proteins can be discussed as well as several have been subject of well designed proteomic an metabolomic studies.
- Have proteomics and/or metabolomics been used to functionally characterize other channels present on the plasma membrane? If so, a section describing the value of both approaches for other channels could be included in this review
- Quantitative proteomic approaches have been applied on cell lines with/without CFTR and on bronchiolar biopsies of CF patients and controls (non-CF patients). I assume both systems have their advantages/disadvantages, which could be discussed. E.g. the cell lines lack the (secondary) inflammatory compound, whereas in the biopsies, proteins involved in inflammation will dominate, but might be physiologically more relevant. Inflammatory reactions in the lungs might also have changed/affected cell phenotype and metabolism in primary cells.
- Most of the alterations in metabolite levels encountered in cells producing an inactive form of CFTR, are probably due to changed extracellular/intracellular ion levels (Cl- and HCO3-)
- The effect of lipid composition of the plasma membrane could be put in perspective. For instance, the activity of the ABC transporter ABCG2 (BCRP) has been shown to heavily depend on membrane cholesterol content
- The mechanism underlying the altered metabolite levels in cells producing mutant CFTR are not discussed in detail
- In the section about metabolomics and biomarker discovery, it is mentioned that in various model systems, many metabolites were found to be affected by (mutant) CFTR (e.g. “32 metabolites differently expressed in CF neonates”. The nature of these metabolites is mostly not given, however. Nor is summarized which metabolites/metabolic pathways are consistently found to be dysregulated in the various studies. These common metabolic pathways might point to the mechanisms underlying the alter metabolism in CF
- In the section about metabolomics and biomarker discovery several studies are mentioned that explored how metabolomics can be used to discover new biomarkers in body fluids and ECB. The authors do however not always mention if these studies yielded anything useful. It would be a good idea the authors in some way evaluate how useful the discovered new biomarkers are with respect to diagnosis, following disease progression etc.
- The potential contribution of altered metabolism to disease progression, if any, is not discussed
- The newly developed metabolic approaches are not compared to the currently standard-of-care test, the pilocarpine-stimulated iontophoresis sweat test, regarding sensitivity/specificity. Maybe a table can be added showing the sensitivity/specificity of the many approaches that have been used, now and in the past, to diagnose CF
- In future directions: Usage of the CF mouse model is not discussed. Have proteomics and/or metabolomics been used to analyze this model system? I would expect the mouse model to provide an excellent tool to study metabolomic alterations induced by CF in vivo as it allows analysis of all major organs affected by CF in a defined system. If CF mice not accurately reflect human CF, the available CF pig model can be considered.
- Some abbreviations are not given in full (e.g. AFT motif). Please check throughout the manuscript.
- Figure 1 seems oversimplified. Arrows only point from the top of the pyramid downwards. However, metabolite levels are known to affect gene expression, e.g. by providing acetyl/methyl groups needed for histone/DNA modifications. Likewise, the metabolite ATP is needed to provide the phosphate groups needed to phosphorylate proteins and gating of the channel.
Minor comments:
- Page 9, lines 371-374: Most screen-positive CF infants result as unaffected carriers or false positive at the sweat test whereas asymptomatic screen-positive with intermediate sweat chloride are classified as CF-screen positive inconclusive diagnosis (CF- SPID) [65]. Difficult to understand what is mean here. Please rephrase.
Author Response
General impression:
The review would be more appealing if more focused on what information on CFTR/CF could be gained from proteomic/metabolomic studies. Specifically, which pressing questions can potentially be answered using proteomics/metabolomics? Lengthy introductions about the various OMICs approaches should be removed. CFTR needs a more extensive introduction. The successful use of proteomics and metabolomics to study other ABC proteins and other ion channels should be discussed, to show how the CFTR field can benefit from these OMICs approaches.
Dear reviewer,
Many thanks for your comments. We have significantly changed the introduction paragraph, focusing more on CFTR rather than on OMICs technologies. We could take into account only some of your suggestions, given the limited space available. Indeed, most of your points would deserve a dedicated review!
Specific comments:
- Unnecessary lengthy introductions are given for genomics, proteomics and metabolomics. Extensive descriptions of sample workup/processing, statistical analysis etc. can be removed. All readers interested in CFTR will know that genes encode proteins and that large parts of the genome consist of non-protein coding regions.
- CFTR deserves a more thorough introduction and should include the fact that CFTR belongs to the ATP-binding cassette proteins and is encoded by ABCC7. The general structure of CTFR should be given. e.g. ATP hydrolyzing domains involved in gating of the channel, the regulatory R domain that can be phosphorylated regulating its open probability etc. Moreover, when discussing the applicability of proteomics and metabolomics to study CFTR, other ABC proteins can be discussed as well as several have been subject of well designed proteomic an metabolomic studies.
Thanks for the suggestions. See also our responses to Rev. 1 and 2. We have shortened down the introduction. We now give a general introduction to CFTR structure with the help of a dedicated figure that also reports the scheme of CFTR post translations modifications.
- Have proteomics and/or metabolomics been used to functionally characterize other channels present on the plasma membrane? If so, a section describing the value of both approaches for other channels could be included in this review:
This is a good suggestion. OMICS have been used to investigate structure and membrane stability of other channels, like large-conductance calcium- and voltage-activated potassium channels (BKCa) and AMPAR receptors. We now mention these studies in our manuscript.
- Quantitative proteomic approaches have been applied on cell lines with/without CFTR and on bronchiolar biopsies of CF patients and controls (non-CF patients). I assume both systems have their advantages/disadvantages, which could be discussed. E.g. the cell lines lack the (secondary) inflammatory compound, whereas in the biopsies, proteins involved in inflammation will dominate, but might be physiologically more relevant. Inflammatory reactions in the lungs might also have changed/affected cell phenotype and metabolism in primary cells.
This is another very interesting point that should indeed deserve a dedicated work.
- Most of the alterations in metabolite levels encountered in cells producing an inactive form of CFTR, are probably due to changed extracellular/intracellular ion levels (Cl- and HCO3-)
- The effect of lipid composition of the plasma membrane could be put in perspective. For instance, the activity of the ABC transporter ABCG2 (BCRP) has been shown to heavily depend on membrane cholesterol content
Thanks for the suggestion. We have added a short discussion on lipids and ABC transporters and, more in general, on lipids and CF.
The mechanism underlying the altered metabolite levels in cells producing mutant CFTR are not discussed in detail.
As above, we now address this point.
- In the section about metabolomics and biomarker discovery, it is mentioned that in various model systems, many metabolites were found to be affected by (mutant) CFTR (e.g. “32 metabolites differently expressed in CF neonates”. The nature of these metabolites is mostly not given, however. Nor is summarized which metabolites/metabolic pathways are consistently found to be dysregulated in the various studies. These common metabolic pathways might point to the mechanisms underlying the alter metabolism in CF
Thanks for the suggestion. We have better clarified these findings.
- In the section about metabolomics and biomarker discovery several studies are mentioned that explored how metabolomics can be used to discover new biomarkers in body fluids and ECB. The authors do however not always mention if these studies yielded anything useful. It would be a good idea the authors in some way evaluate how useful the discovered new biomarkers are with respect to diagnosis, following disease progression etc.
- The potential contribution of altered metabolism to disease progression, if any, is not discussed
- The newly developed metabolic approaches are not compared to the currently standard-of-care test, the pilocarpine-stimulated iontophoresis sweat test, regarding sensitivity/specificity. Maybe a table can be added showing the sensitivity/specificity of the many approaches that have been used, now and in the past, to diagnose CF
- In future directions: Usage of the CF mouse model is not discussed. Have proteomics and/or metabolomics been used to analyze this model system? I would expect the mouse model to provide an excellent tool to study metabolomic alterations induced by CF in vivo as it allows analysis of all major organs affected by CF in a defined system. If CF mice not accurately reflect human CF, the available CF pig model can be considered.
This is an interesting comment. Very few omics experiments were done on animal CF models, but we now mention this point. We are now citing our own work on bile acids in mice stool.
- Some abbreviations are not given in full (e.g. AFT motif). Please check throughout the manuscript.
- Figure 1 seems oversimplified. Arrows only point from the top of the pyramid downwards. However, metabolite levels are known to affect gene expression, e.g. by providing acetyl/methyl groups needed for histone/DNA modifications. Likewise, the metabolite ATP is needed to provide the phosphate groups needed to phosphorylate proteins and gating of the channel.
Thanks for your comments. We have revised all the figures of our manuscript and we have added a figure illustrating CFTR structure at the plasma membrane, indicating the ATP role in the gating.
Minor comments:
- Page 9, lines 371-374: Most screen-positive CF infants result as unaffected carriers or false positive at the sweat test whereas asymptomatic screen-positive with intermediate sweat chloride are classified as CF-screen positive inconclusive diagnosis (CF- SPID) [65]. Difficult to understand what is mean here. Please rephrase.
Thanks. We have now removed this part.
Reviewer 4 Report
The authors attempt to review the most up-to-date information on proteomic and metabolomic studies in cystic fibrosis (CF). I commend the authors for undertaking the challenge. The review could be informative and useful for the research community after a major revision, including focusing on the subject being reviewed and correcting the English language.
Here are the major points that need to be addressed:
The introduction is too long and contains information covered already by several excellent reviews. The manuscript should be an expert review on CF related proteomics and metabolomics. The scientific community already knows proteomics and metabolomics and the long introduction is unnecessary.
The comparison or proteomics and metabolomics workflows is very long, confusing, and it is unclear to me why the workflows are being compared to each other. The authors need to succinctly introduce the concepts and discuss proteomics and metabolomics of CF.
The conclusion and future perspectives section is one page-long and completely unfocussed. It should very briefly, in no more than a paragraph, list the major points reviewed in the manuscript.
The abstract needs to be re-written after the revision to briefly signal what would be reviewed. It is very unfocussed and reading the abstract does not inform what the reader would find in the manuscript.
The discussion of CF pathophysiology is fragmentary and not up to par with current understanding of the disease. The lack of depth discourages from reading the manuscript.
The text reads more like an essay. It is not divided into paragraphs. The authors should break the text after finishing discussion of a group of studies or a concept.
Language needs major work. The entire manuscript requires revision to correct the syntax. The authors should avoid repetitions and long, confusing sentences. They should avoid speculations and adhere more to the review of published data.
Below are few examples of the confusion created by language and few suggestions. The authors should not take this as if only these examples would need correction.
Abstract: The last sentence, phrase after the come is vague and should be removed.
Introduction:
Line 29: the word “both” should be removed as if falsely suggests that Cl and HCO3 are conducted by CFTR at the same time.
Line 29: change “epithelial surfaces” to “epithelium”
Line 32: The statement that “viscous mucus is life threatening” should be rephrased to provide precise patho-mechanism of cystic fibrosis lung disease.
Line 36: The authors have to rephrase the sentence stating that Ms. Andersen has been studying CF since 1930.
Line 41 What “papers about CF” are?
Proteomics and metabolomics workflows:
Line 112: The meaning of the following sentence is very confusion “Proteomics and metabolomics workflows significantly differ in most of the steps of their workflows, represented and summarized in Figure 1, Panel B.”
Line 132: Unclear meaning of the following sentence: “The same occurs, off course, for targeted metabolomics experiments.”
Line 487: Unclear: “We also realized how few the published omics studies focused on CFTR mutations other than F508del are.”
Author Response
Comments and Suggestions for Authors
The authors attempt to review the most up-to-date information on proteomic and metabolomic studies in cystic fibrosis (CF). I commend the authors for undertaking the challenge. The review could be informative and useful for the research community after a major revision, including focusing on the subject being reviewed and correcting the English language.
Here are the major points that need to be addressed:
The introduction is too long and contains information covered already by several excellent reviews. The manuscript should be an expert review on CF related proteomics and metabolomics. The scientific community already knows proteomics and metabolomics and the long introduction is unnecessary. The comparison or proteomics and metabolomics workflows is very long, confusing, and it is unclear to me why the workflows are being compared to each other. The authors need to succinctly introduce the concepts and discuss proteomics and metabolomics of CF.
Thanks. We have shortened the introduction part.
The conclusion and future perspectives section is one page-long and completely unfocussed. It should very briefly, in no more than a paragraph, list the major points reviewed in the manuscript.
Thanks. We also revised the conclusion of our Review.
The abstract needs to be re-written after the revision to briefly signal what would be reviewed. It is very unfocussed and reading the abstract does not inform what the reader would find in the manuscript. The discussion of CF pathophysiology is fragmentary and not up to par with current understanding of the disease. The lack of depth discourages from reading the manuscript.
Thanks. While the focus of this work remains a general overview on the role of omics, we have improved the introduction of CFTR structure and biology.
The text reads more like an essay. It is not divided into paragraphs. The authors should break the text after finishing discussion of a group of studies or a concept.
Thanks. Done.
Language needs major work. The entire manuscript requires revision to correct the syntax. The authors should avoid repetitions and long, confusing sentences. They should avoid speculations and adhere more to the review of published data.
Below are few examples of the confusion created by language and few suggestions. The authors should not take this as if only these examples would need correction.
Abstract:
The last sentence, phrase after the come is vague and should be removed.
Introduction:
Line 29: the word “both” should be removed as if falsely suggests that Cl and HCO3 are conducted by CFTR at the same time.
Line 29: change “epithelial surfaces” to “epithelium”
Line 32: The statement that “viscous mucus is life threatening” should be rephrased to provide precise patho-mechanism of cystic fibrosis lung disease.
Line 36: The authors have to rephrase the sentence stating that Ms. Andersen has been studying CF since 1930.
Line 41 What “papers about CF” are?
Proteomics and metabolomics workflows:
Line 112: The meaning of the following sentence is very confusion “Proteomics and metabolomics workflows significantly differ in most of the steps of their workflows, represented and summarized in Figure 1, Panel B.”
Line 132: Unclear meaning of the following sentence: “The same occurs, off course, for targeted metabolomics experiments.”
Line 487: Unclear: “We also realized how few the published omics studies focused on CFTR mutations other than F508del are.”
Thanks we have made changes to the text to fix these points.
Round 2
Reviewer 3 Report
General:
Needs input from a native speaker to correct errors in grammar and improve readability. Introductions for proteomics and metabolomics are still lengthy. It would be better to explain in more detail what both omics approaches have taught us about CFTR biology. The latter is not always very clear in the current manuscript. For instance, on page 11, lines 376-381: “Moreover, they demonstrated that the misfolded F508del-CFTR shows a replacement of methylation(methyl groups?) with (by?) ubiquitination (ubiquitin groups?) sites. This study is particularly relevant, because it is the first underlining the essential role of PMTs;” Essential role in what? I assume in degradation of misfolded CFTR, but this is not at all clear. The authors should carefully go over their manuscript to formulate more accurately/precisely.
Specific points:
- Introduction
- It is nowhere mentioned that ABCC7 encodes CFTR.
- It is nowhere indicated that CFTR is an ATP-gated ion channel
- Proteomics
- Too much general information about genomics, proteomics and metabolomics.
- Description of proteomics/metabolomics is full of clichés, like “Despite many advanced methods developed, we are still far from achieving the total coverage of the genome”
4.1. MudPIT-based interactomics
- Is the different interactome caused by the mutation or by retention in the ER? Have there been efforts to identify the cause of the different interactome between wt and F508del CFTR, for instance by growing the cells at a lower temperature? Do chemical/pharmacological chaperones change the F508del mutant into the direction of the wt CFTR interactome?
- Global changes in protein expression observed in CF
- Instead of listing how many proteins were found to be differentially present in cells producing wild type CFTR versus cells that produced the F508del variant, it would be better to explain what was learned about CFTR biology from these experiments. For instance, which pathways directly related to CF were altered?
- Metabolomics and CFTR research.
- Metabolomics identified several pathways that were altered in CF human airway epithelial cells. Was is cause and what is consequence?
- Using metabolomics to follow treatment effect is interesting, but should be better explained. It is, for instance not clear if the serum metabolite profile in patients treated with Lumacaftor/Ivacaftor changed towards that of healthy individuals. I guess it did, but should be more clearly explained. Which serum metabolite changes are found in untreated CF patients and how do these change after treatment?
- Some explanation is needed why amitriptyline and fenretinid can be used to treat CF. What is the mode of action of these drugs? What is their biological target?
Author Response
Reviewer 1
Comments and Suggestions for Authors
General:
Needs input from a native speaker to correct errors in grammar and improve readability. Introductions for proteomics and metabolomics are still lengthy. It would be better to explain in more detail what both omics approaches have taught us about CFTR biology. The latter is not always very clear in the current manuscript. For instance, on page 11, lines 376-381: “Moreover, they demonstrated that the misfolded F508del-CFTR shows a replacement of methylation(methyl groups?) with (by?) ubiquitination (ubiquitin groups?) sites. This study is particularly relevant, because it is the first underlining the essential role of PMTs;” Essential role in what? I assume in degradation of misfolded CFTR, but this is not at all clear. The authors should carefully go over their manuscript to formulate more accurately/precisely.
Thank you very much for your comments and suggestions. We made the changes you suggested in order to improve the manuscript. Please find below our point-to-point responses to your comments.
Specific points:
- Introduction
- It is nowhere mentioned that ABCC7encodes CFTR.
Thanks. We added this part.
- It is nowhere indicated that CFTR is an ATP-gated ion channel.
In the introduction, we have already written how CFTR works, including that it needs ATP for its gating (line 35, line 43).
- Proteomics
- Too much general information about genomics, proteomics and metabolomics.
We do not agree with you. We have already reduced the long introduction to these fields. We think that general information about genomics, proteomics and metabolomics are useful to the reader.
- Description of proteomics/metabolomics is full of clichés, like “Despite many advanced methods developed, we are still far from achieving the total coverage of the genome”.
Many thanks for the suggestion.
4.1. MudPIT-based interactomics
- Is the different interactome caused by the mutation or by retention in the ER? Have there been efforts to identify the cause of the different interactome between wt and F508del CFTR, for instance by growing the cells at a lower temperature? Do chemical/pharmacological chaperones change the F508del mutant into the direction of the wt CFTR interactome?
Thank you for your comments. We clarified all these points.
- Global changes in protein expression observed in CF
- Instead of listing how many proteins were found to be differentially present in cells producing wild type CFTR versus cells that produced the F508del variant, it would be better to explain what was learned about CFTR biology from these experiments. For instance, which pathways directly related to CF were altered?
As said for the previous point, we added clarifications.
- Metabolomics and CFTR research.
- Metabolomics identified several pathways that were altered in CF human airway epithelial cells. Was is cause and what is consequence?
We reported metabolomic studies that highlighted changes at metabolomic level after some treatments. For each study we mentioned, we described the metabolic alterations detected, which are due to the treatment.
- Using metabolomics to follow treatment effect is interesting, but should be better explained. It is, for instance not clear if the serum metabolite profile in patients treated with Lumacaftor/Ivacaftor changed towards that of healthy individuals. I guess it did, but should be more clearly explained. Which serum metabolite changes are found in untreated CF patients and how do these change after treatment?
The reported study did not address these points. The Authors used metabolomics to monitor differences before/after the treatment. The comparison with healthy subjects was not the final aim of this study.
- Some explanation is needed why amitriptyline and fenretinid can be used to treat CF. What is the mode of action of these drugs? What is their biological target?
We added the description of these drugs.

Reviewer 4 Report
Content:
In the Introduction, the authors decided to add large section describing the CFTR channel domains and classes of CFTR gene mutations but failed to describe the CF pathophysiology. It is critical to provide the reader with a focused background that sets the stage for the review. At this time, the authors did not achieve the balance. Some of the extended descriptions should be moved to the main body of the review manuscript to provide rationale for the studies.
Language:
The English language is still substandard and lacks the scientific depth and rigor. There are many long sentences with incorrect syntax and exaggerations. Examples are listed below:
Line 56 – “The impaired activity of the channel alters the salt homeostasis, leading to defective mucus clearance.” The authors inform and CFTR conducts chloride and bicarbonate transport. How does it lead to salt homeostasis and where?
Line 57 – “The presence of viscous mucus is particularly life-threatening in the respiratory tract, because it causes recurrent lung infections and chronic inflammation.”
The above sentence is poorly constructed and ambiguous. It would confuse the reader.
The following sentences are examples of major problems with the English language that have to be addressed by a professional English translator. Only few examples are as follows:
Line 63: “Despite the first study on cystic fibrosis was published in the 1930s by Dorothy Hansine Andersen [9], the CFTR gene was identified and cloned only in 1989 [10].”
Line 65: “Since its discovery, CFTR has been widely studied both for its structure and its function.”
Line 70: “In this respect, CF researchers now benefit from the most advanced techniques to investigate the biological space with high coverage, great details, high sensitivity and, more importantly, at a global level: the omics.
Line 72: “Omics represent a set of biomolecular disciplines, rapidly developed in the last 20 years, which allowed to dramatically increase our understanding of the architecture and functioning of biological systems using large-scale molecular-level measurements, from DNA up to metabolic reactions.”
The very long sentence that poorly introduces ‘omics’.
Line 94: “While the DNA of a cell is essentially static in time, protein and metabolite content is dynamic and it varies from time to time, from tissue to tissue and in different physical or pathological conditions.”
It is incorrectly to state that ‘DNA is static in time’. Similar to proteins and metabolites, DNA undergoes very active regulation. It is unclear what the authors mean by saying that ‘protein and metabolite content varies from time to time’. This is not a language used in science.
Line 97: “ The pathway that leads from DNA to protein synthesis and metabolites processing, involves numerous biochemical processes controlled at each step of the path. It is well known that metabolites play an essential role in gene regulation, a striking example being the role of cholesterol [12] or 99 glucose [13].
In the same sentence, the authors state that the role of metabolites is well known but call their role in cholesterol regulation a ‘striking’ example.
Similar examples of major language problems are present throughout the entire manuscript and significantly diminish the potential value of the review.
Line 550: “The composition of the lung microbiome in CF patients has been widely investigated for many different aims, including the discrimination between CF and non-CF patients, antimicrobial clinical trials, and the evaluation of multi-drug resistance.“
This is yet another of many examples where the information provided by the authors is inconsistent with the current state of knowledge and practice. Lung microbiome is not used to diagnose CF patients.
Very confusing figure legend:
“Figure 1. Panel A: representation of the investigation pathway from the genotype to the phenotype.”
Design of the manuscript:
The manuscript design makes the review difficult to follow and should be improved. The general information on proteomics and metabolomics (currently section 2 and 3) should be combined, shortened, and moved to the Introduction. The authors should focus the review of the techniques to studies of CFTR, as stated in the title. Section 4 is in fact the main subject of the review manuscript.
Description of the proteomics approaches (4.1 and 4.2) should be combined into one subsection.
The in
Sub-headings:
The authors change the style used for the subheadings that breaks the flow of the manuscript. Subheadings should be precise and informative. Understanding many of the vague sub-headings is possible only after reading the text that follows. For example:
“9. Omics and the altered CF microbiome.”
Is there an altered and non-altered CF microbiome?
I suggest the following change: Omics to study the CF microbiome
“4.3 Interactomics at AFT motif” should be changed to Elucidating the AFT-based CFTR interactome.
These are just few examples and the authors should take similar approach to modify other sub-headings.
4.5 It would be unclear to the general reader why the authors discuss the omics of the G551D-CFTR. The authors do not provide the rationale.
Author Response
Reviewer 2:
Comments and Suggestions for Authors
Content:
In the Introduction, the authors decided to add large section describing the CFTR channel domains and classes of CFTR gene mutations but failed to describe the CF pathophysiology. It is critical to provide the reader with a focused background that sets the stage for the review. At this time, the authors did not achieve the balance. Some of the extended descriptions should be moved to the main body of the review manuscript to provide rationale for the studies.
Language:
The English language is still substandard and lacks the scientific depth and rigor. There are many long sentences with incorrect syntax and exaggerations. Examples are listed below:
Line 56 – “The impaired activity of the channel alters the salt homeostasis, leading to defective mucus clearance.” The authors inform and CFTR conducts chloride and bicarbonate transport. How does it lead to salt homeostasis and where?
Line 57 – “The presence of viscous mucus is particularly life-threatening in the respiratory tract, because it causes recurrent lung infections and chronic inflammation.”
The above sentence is poorly constructed and ambiguous. It would confuse the reader.
Many thanks for the suggestion. We changed these parts
The following sentences are examples of major problems with the English language that have to be addressed by a professional English translator. Only few examples are as follows:
Line 63: “Despite the first study on cystic fibrosis was published in the 1930s by Dorothy Hansine Andersen [9], the CFTR gene was identified and cloned only in 1989 [10].”
Line 65: “Since its discovery, CFTR has been widely studied both for its structure and its function.”
Line 70: “In this respect, CF researchers now benefit from the most advanced techniques to investigate the biological space with high coverage, great details, high sensitivity and, more importantly, at a global level: the omics.”
Line 72: “Omics represent a set of biomolecular disciplines, rapidly developed in the last 20 years, which allowed to dramatically increase our understanding of the architecture and functioning of biological systems using large-scale molecular-level measurements, from DNA up to metabolic reactions.”
The very long sentence that poorly introduces ‘omics’.
Thank you very much for giving us some examples. We changed the sentences to improve the manuscript.
Line 94: “While the DNA of a cell is essentially static in time, protein and metabolite content is dynamic and it varies from time to time, from tissue to tissue and in different physical or pathological conditions.”
It is incorrectly to state that ‘DNA is static in time’. Similar to proteins and metabolites, DNA undergoes very active regulation. It is unclear what the authors mean by saying that ‘protein and metabolite content varies from time to time’. This is not a language used in science.
Line 97: “ The pathway that leads from DNA to protein synthesis and metabolites processing, involves numerous biochemical processes controlled at each step of the path. It is well known that metabolites play an essential role in gene regulation, a striking example being the role of cholesterol [12] or 99 glucose [13].
In the same sentence, the authors state that the role of metabolites is well known but call their role in cholesterol regulation a ‘striking’ example.
Similar examples of major language problems are present throughout the entire manuscript and significantly diminish the potential value of the review.
Thanks for your comments. We changed the sentences.
Line 550: “The composition of the lung microbiome in CF patients has been widely investigated for many different aims, including the discrimination between CF and non-CF patients, antimicrobial clinical trials, and the evaluation of multi-drug resistance.“
This is yet another of many examples where the information provided by the authors is inconsistent with the current state of knowledge and practice. Lung microbiome is not used to diagnose CF patients.
Very confusing figure legend:
“Figure 1. Panel A: representation of the investigation pathway from the genotype to the phenotype.”
We changed the figure legend.
Design of the manuscript:
The manuscript design makes the review difficult to follow and should be improved. The general information on proteomics and metabolomics (currently section 2 and 3) should be combined, shortened, and moved to the Introduction. The authors should focus the review of the techniques to studies of CFTR, as stated in the title. Section 4 is in fact the main subject of the review manuscript.
Description of the proteomics approaches (4.1 and 4.2) should be combined into one subsection.
Thank you for this comment but we decided not to combine the two proteomics approaches. They are quite different and describing them separately allows us to highlight some points useful to the reader.
The in
Sub-headings:
The authors change the style used for the subheadings that breaks the flow of the manuscript. Subheadings should be precise and informative. Understanding many of the vague sub-headings is possible only after reading the text that follows. For example:
“9. Omics and the altered CF microbiome.”
Is there an altered and non-altered CF microbiome?
I suggest the following change: Omics to study the CF microbiome
Many thanks for the suggestion. We changed the sub-heading.
“4.3 Interactomics at AFT motif” should be changed to Elucidating the AFT-based CFTR interactome.
These are just few examples and the authors should take similar approach to modify other sub-headings.
Many thanks for the suggestion. We changed the sub-heading.
4.5 It would be unclear to the general reader why the authors discuss the omics of the G551D-CFTR. The authors do not provide the rationale.
Thank you for this comment. We added some explanations to this part.
Round 3
Reviewer 3 Report
The readability has dramatically improved.
Reviewer 4 Report
The authors adequately responded to my suggestions.